# Improving Saline–Alkaline Soil and Ryegrass Growth with Coal Gangue Treatments

**DOI:** 10.3390/plants13233419

**Published:** 2024-12-05

**Authors:** Siqi Li, Xingqiang Li, Xiaolin Qiang, Zhao Yu, Hongyuan Li, Zhaojun Sun, Qian Li, Jun He, Lei Han, Ningxin Zhao

**Affiliations:** 1School of Civil and Hydraulic Engineering, Ningxia University, Yinchuan 750021, China; siqi7li@163.com (S.L.); 12021140101@stu.nxu.edu.cn (X.L.); qiangxiaolin123@163.com (X.Q.); nanxia_yu@163.com (Z.Y.); lhy2022666@163.com (H.L.); 2School of Geography and Planning, Ningxia University, Yinchuan 750021, China; li_q@nxu.edu.cn (Q.L.); hejun3025@163.com (J.H.); layhan@163.com (L.H.); 3Guoneng Jiangsu Jianbi Power Generation Co., Ltd., Nantong 226000, China; 18209256534@163.com

**Keywords:** solid waste, cover, soil amelioration, principal component analysis

## Abstract

Soil covering is a soil management technique used to address issues such as high alkalinity, nutrient deficiency, and a low soil and water-holding capacity in saline–alkali lands. Coal gangue, a solid waste generated from coal mining, contains nutrients and trace elements comparable to those in soil, making it a promising covering material. This study investigates the effects and mechanisms of coal gangue coverage on the physicochemical properties of soil, enzyme activity, and the growth of ryegrass. The experimental treatments comprised (1) three coal gangue particle sizes (0–0.5 cm, 0.5–1 cm, and 1–2 cm) and (2) three coverage thicknesses (4 cm, 8 cm, and 12 cm). The results show that with particle sizes of 0–1 cm and cover thicknesses of 8–12 cm, the saturated water content of the soil increased by 11.1% to 17.23%, the content of organic matter rose by 4.99 to 13.64 mg/kg, the total nitrogen increased by 0.07 to 0.12 mg/kg, and the urease activity increased by 0.56 to 0.64 μg/(d·g). Our analyses indicated that an improvement in soil porosity, a reduction in pH, and an increase in urease, organic matter, and total nitrogen collectively promoted ryegrass growth, with similar impacts. Among the treatments, coverage with 12 cm of coal gangue with a size of 0.5–1 cm produced the best results, increasing the ryegrass height by 16.24 cm, primarily due to a high level of soil porosity and elevated urease activity. In conclusion, coal gangue coverage significantly improves the properties of soil, enhances enzyme activity, and promotes ryegrass growth.

## 1. Introduction

Mulching is an essential soil management technique used globally to control temperature and moisture, prevent soil erosion, and enhance crop yield [1]. Common mulch types include straw, horticultural ground cloth, and gravel [2]. Gravel mulching is notable for its capacity to boost the water content of the soil, elevate its organic matter content, improve its fertility, enhance its microbial activity, and promote crop growth and development, making it a topic of widespread interest [3]. For example, Li Wangcheng et al. conducted research on the environmental impacts of gravel-covered farmland, emphasizing the effects of gravel cover on the hydrothermal conditions, salinity, organic matter content, and microbial and enzyme activities of soil, as well as soil and water retention mechanisms [4]. The production of significant amounts of solid waste such as coal gangue from coal mining and processing presents challenges, resulting in coal gangue mountains that occupy land resources and are susceptible to spontaneous combustion. Therefore, the effective utilization of coal gangue is a pressing issue [5,6,7]. China’s coal gangue resembles gravel in its granular form, containing mineral elements similar to soil and rich in carbonaceous shale. Notably, it has an organic matter content of 15–25% and has an abundance of essential growth elements such as boron, zinc, copper, cobalt, molybdenum, manganese, and other trace elements, surpassing the content in soil by 2–10 times and exhibiting strong absorption capabilities [8,9,10]. Consequently, gangue serves as an ideal material for the amendment of soil, improving the quality of soil, and promoting crop growth in arid regions.

Various studies have shown that gangue cover with different thicknesses and particle sizes can boost soil water infiltration, reduce evaporation, enhance soil moisture levels, and alleviate problems such as crusting, sanding, and salinization in soil [11,12]. Additionally, the presence of humic acid, organic matter, silica, potassium, iron, and various rare elements in gangue cover significantly affects the pH and nutrient content of soil due to interactive effects related to the cover thickness, particle size, and composition [13]. Some studies have assessed the physicochemical properties of soil through alfalfa potting tests, revealing that gangue mulching can enhance the capacity of soil to retain water and fertilizer while promoting plant growth [14,15]. However, comprehensive evaluations of the impacts of gangue cover on the physical properties of soil, its chemical properties, enzyme activities, and crop growth and development remain scarce. Further research on gangue’s effects on soil nutrients, enzyme activities, and crop growth mechanisms is needed to deepen our understanding of this area of study.

Furthermore, saline and alkaline land covers approximately 9.5438 × 10^8^ hm^2^ globally, with China accounting for approximately 9.913 × 10^7^ hm^2^ (10.39% of the world’s total); this is primarily concentrated in the northwest, north, northeast, and coastal regions of China. These areas are marked by excessive soil salinity and alkalinity, posing challenges to crop growth and jeopardizing agricultural productivity and ecological balance [16,17,18,19,20]. Enhancing saline and alkaline lands can not only boost the productivity of land and enhance crop yields, but it can also have significant implications for national food security and sustainable economic growth [21]. Research on coal gangue as a mulching technology is limited, despite its potential to improve the physical and chemical properties of saline soils. Further research is needed to understand its effects on the characteristics of soil and crop growth in saline environments.

This study investigates the use of coal gangue from the Ningdong Energy and Chemical Base as an experimental material. The heavy metal content in the coal gangue has been analyzed, confirming that all indicators meet the established screening values. This research focuses on ryegrass (*Lolium perenne*), which is characterized by rapid growth, substantial biomass, and strong adaptability, cultivated in sandy loam soil with high salinity and alkalinity. Ryegrass is known for its rapid germination, tolerance to adverse conditions, resistance to salinity and alkalinity, and robustness against pests and wind, making it an ideal candidate for soil improvement and phytoremediation studies in the saline–alkaline soils of arid regions. Our hypothesis posits that the application of coal gangue as a surface cover in potting soil positively influences the properties of soil, its enzyme activity, and crop growth. The collected data will be analyzed to elucidate the underlying mechanisms of these changes and to establish optimal design parameters for coal gangue coverage.

## 2. Results

### 2.1. The Impact of Coal Gangue Cover on Soil Physical Characteristics

Soil moisture is vital for ecosystem health, and inadequate moisture leads to drought; this limits the volume of water available to plants and soil organisms, and negatively impacts growth and the overall health of soil [22]. The water-holding capacity (WHC) and saturated water content (WFPS) of the soil significantly increased with the coal gangue cover thickness, surpassing the control group (CK), as shown in Figure 1. For particle sizes S1 (0–0.5 cm), S2 (0.5–1 cm), and S3 (1–2 cm), the WHC for T2 (8 cm) rose by 2.65%, 1.88%, and 2.19% compared to T1 (4 cm), while T3 (12 cm) increased by 3.59%, 2.31%, and 2.44%. The WFPS for T2 increased by 9.41%, 9.21%, and 9.44% compared to T1, while T3 increased by 13.74%, 12.57%, and 24.54%. This demonstrates that the coal gangue cover effectively retains moisture and reduces evaporation [23].

The optimal soil bulk density for crop growth is between 1.14 and 1.30 g/cm^3^ [24]. In this experiment, increasing the gangue cover thickness initially decreased the soil bulk density (SBD) before it rose again. The control had an SBD of 1.54 g/cm^3^, indicating excessive compaction. At T1 and T2, the SBD significantly decreased in treatments S1, S2, and S3 compared to CK, with reductions of 0.03, 0.12, and 0.13 g/cm^3^ at T1 and 0.12, 0.18, and 0.19 g/cm^3^ at T2. However, the SBD significantly increased at T3 compared to CK. The bulk density of soil affects its permeability, water retention, nutrient capacity, and erosion resistance [25]. Coal gangue reduces topsoil erosion and the hardening caused by sun exposure, preserving the structure of soil and enhancing infiltration through larger pore spaces. While coarser gangue particles reduce compaction in smaller amounts, their excessive use can lead to increased compaction due to their higher density.

### 2.2. The Impact of Coal Gangue Cover on Soil Chemical Properties

The correlation between gangue cover thickness and soil pH on day 70 is shown in Table 1. The pH decreased as the thickness increased for the same particle size. At S1, T2 reduced the pH by 0.12 and T3 reduced the pH by 0.24 compared to T1; for S2, T2 decreased the pH by 0.12 and T3 reduced the pH by 0.15; and for S3, T2 decreased the pH by 0.16 and T3 reduced the pH by 0.19. Significant differences in pH between the control and treatments appeared at T2, with average decreases of 0.39, 0.51, 0.36, 0.34, and 0.37 for S1T2, S1T3, S2T3, S3T2, and S3T3. Under T1, the differences between grain sizes were insignificant, while significant differences appeared between S1 and S2 under T2 and T3. Untreated soil had an organic matter content of 7.9 mg·kg^−1^ and total nitrogen of 0.12 mg·kg^−1^, indicating low nutrient levels [26]. Soil organic matter (SOM) increased with a greater gangue cover thickness. Under S1, SOM rose by 1.42, 1.7, and 2.72-fold for T1, T2, and T3 compared to the control (CK). For S2, these values increased by 1.38, 1.63, and 2.48, and for S3, they increased by 1.13, 1.49, and 1.94. The total nitrogen (TN) content also increased with cover thickness. For S1, TN rose by 0.05, 0.07, and 0.10 mg/kg at T1, T2, and T3 compared to CK. S2 showed increases of 0.04, 0.08, and 0.12 mg/kg, while S3 increased by 0.03, 0.05, and 0.09 mg/kg, with the largest increase being observed from T2 to T3. Both TN and organic matter significantly increased from T2, indicating that the cover thickness primarily affected these properties.

The total salt content in soil with gangue cover was lower than that in the control group (CK), with 1.69, 1.75, 1.93, and 1.8-fold reductions under treatments S1, S2, and S3, respectively, as shown in Figure 2. T1 had a slightly lower salt content than T2, at 0.24 g·kg^−1^. T1 and T2 exhibited significantly lower salt levels at the same grain size, with T1 showing a greater reduction. T3 exhibited no significant reduction, indicating that the cover thickness influenced salt reduction. Gangue cover effectively reduced the total salt content, with smaller reductions being observed for thicker covers and the particle size having a minimal impact. It also slowed the movement of water and enhanced infiltration [27]. The highest soil alkalinity (ESP), at 24.65%, was found when cover was not used. Reductions were more pronounced with smaller particle sizes; T2 decreased the ESP by 5.26% and T3 decreased the ESP by 6.93% at S1. Increasing the particle size slightly elevated the ESP. The highest effective phosphorus (AP) content, at 4.1 mg/kg, was found when the cover was not used and was significantly lower than that in the covered treatments. At S1, T1, T2, and T3 decreased the AP by 3.43 mg/kg, 3.37 mg/kg, and 3.24 mg/kg, respectively, compared to CK. The AP content of the soil increased with larger grain sizes, with the lowest mean AP of 0.67 mg/kg being observed in S1T1. The fast-acting potassium (AK) content was 48.64 mg/kg when the cover was not used, but reached 79.93 mg/kg with T3 and S3; this is 1.64 times higher than CK. Increasing the cover thickness significantly boosted the AK content of the soil. At the same thickness, the AK content initially decreased and then increased with increasing grain size.

In conclusion, applying a coal gangue cover effectively reduced the accumulation of surface salt and alkalinity while increasing the effective phosphorus content and alleviating secondary salinization. Changes in the cover thickness and particle size had a minimal impact on the effective phosphorus content, likely due to the gangue’s consistent adsorption capacity.

### 2.3. The Impact of Coal Gangue Cover on Soil Enzyme Activity

The selection of soil urease, soil alkaline phosphatase, and soil sucrase as representative enzymes involved in the biological cycles of nitrogen, phosphorus, and carbon metabolism, respectively, is shown in Figure 3. These enzymes were chosen to investigate the impacts of different gangue cover thicknesses and particle sizes on the enzyme activities of the soil [28]. The figure shows that the activities of the three enzymes in all treatments were significantly higher than those in the control group (CK). This suggests that gangue mulching can supply essential nutrients to soil microorganisms, stimulate microbial growth, and enhance the enzyme activities of soil [29].

The soil urease (URE) activity rose as the cover thickness increased. Under S1, S2, and S3, T2 showed increases of 11.65, 12.28, and 26.94 μg/(d·g), while T3 showed increases of 33.91, 47.95, and 79.41 μg/(d·g) compared to T1. The increase in URE activity at T3 was significant, reaching up to 1.87 times that of CK, with the highest activity being observed in S2T2. The alkaline phosphatase (ALP) activity in the also increased as the thickness of the mulch increased. T2 increased by 0.09 to 0.04 μmol/(d·g) compared to T1, while T3 showed minimal changes. The ALP activity increased by up to 2.14 times compared to CK. The sucrase (INV) activity of the soil increased as the cover thickness increased: T2 increased by 0.04 to 0.4 mg/(d·g), and T3 increased by significant amounts compared to T1. The INV activity at T3 increased by 2.84 to 2.43 times compared to CK, with the highest value being observed in S1T3. Overall, the enzyme activity of the soil significantly responded to gangue cover, influenced by factors such as moisture, temperature, pH, and microbial activity [30].

### 2.4. The Impact of Coal Gangue Cover on Ryegrass Plant Height and Biomass

The variations in the plant height and biomass of ryegrass, depicted in Figure 4, serve as quantitative indicators of its current state of growth and development. Additionally, they reflect the physical expressions of the environmental conditions necessary for its optimal growth, including the physicochemical properties and enzyme activities of the soil [31]. This study utilized a logistic growth model to describe the growth dynamics of ryegrass plant height (H). By fitting the ryegrass plant height data, the parameter values and eigenvalues of the logistic model were determined under various gangue cover conditions. The coefficient of determination (R^2^) for each treatment group in the logistic model exceeded 0.94, indicating a good fit, with slopes consistently higher than those of the CK group, as shown in the figure. Notably, for particle size S1, parameter A reached a maximum of 1.2141 at cover thickness T3, with parameter B initially decreasing before increasing with the gangue cover thickness. Similarly, for particle sizes S2 and S3, parameter A exhibited a similar trend as the cover thickness varied, peaking at 1.3005 for T3; meanwhile, parameter B displayed a corresponding decrease, also reaching a peak at T3. These observations align with the actual growth patterns of ryegrass, confirming the suitability of the logistic growth model for assessing the plant height of ryegrass [31].

Fresh weight and dry weight are essential metrics for evaluating the growth and accumulation of dry matter in ryegrass. These indicators reflect the growth status and water content of ryegrass, indicating its freshness [32]. The above-ground fresh weight (AFW) of ryegrass under cover thicknesses T2 and T3 significantly exceeded that of the control group (CK) for gangue particle sizes S1, S2, and S3. Specifically, the AFW for S1T2, S1T3, S2T2, S2T3, S3T2, and S3T3 increased by 53.6 g, 58.7 g, 69 g, 78.2 g, 20.8 g, and 38.6 g, respectively, compared to the CK, as shown in Figure 5. Similarly, the underground fresh weight (UFW) of ryegrass showed significant increases under the T2 and T3 cover thicknesses relative to CK for gangue sizes S1 and S2. Specifically, S2T2, S2T3, S3T2, and S3T3 increased by 20.12 g, 23.41 g, 29.99 g, and 35.74 g, respectively, compared to CK. This observation suggests that cover thicknesses exceeding 8 cm for smaller gangue particles positively influenced both the above-ground and below-ground fresh weight of ryegrass, with optimal effects observed at S2T3. Notably, the ryegrass root length was significantly lower under the T1 cover thickness compared to CK, and the growth rate decreased under the T3 cover thickness; this implies that minimal gangue cover thicknesses had a limited impact on soil enhancement and the supply of nutrients to ryegrass, while excessive coverage inhibited root growth. Excessive gangue deposition on the root surface obstructed the epidermal pores of the roots, impeding water and nutrient absorption and thereby inhibiting the growth of the ryegrass root system [33].

The above-ground dry weight (ADW) and below-ground dry weight (UDW) of ryegrass showed similar trends under various gangue cover treatments, consistent with the patterns observed in above-ground fresh weight (AFW) and below-ground fresh weight (UFW). For gangue particle sizes S1, S2, and S3, the ADW of ryegrass under cover thicknesses T2 and T3 significantly exceeded that of the control group (CK), with S1T2, S1T3, S2T2, S2T3, S3T2, and S3T3 increasing by 10.3 g, 11.5 g, 13.5 g, 15.3 g, 3.8 g, and 7.3 g relative to the CK. The UDW of ryegrass notably surpassed that of CK under the T2 and T3 cover thicknesses for gangue sizes S1, S2, and S3, with S1T2, S1T3, S2T2, S2T3, S3T2, and S3T3 showing increases of 5.67 g, 6.47 g, 7.59 g, 8.6 g, 1.62 g, and 3.45 g compared to CK. These results suggest that fine-grained gangue mulch with cover thicknesses exceeding 8 cm positively influenced both the above-ground and below-ground dry weight of ryegrass, with optimal effects observed at S2T3.

### 2.5. Effects of Different Coal Gangue Coverages on Soil pH and Ryegrass Plant Height

We then investigated the effect of changes in the soil pH under different coal gangue cover treatments on the ryegrass height. Except for the initial growth period (within 14 days), changes in the coal gangue coverage thickness and particle size had little effect on the height of the ryegrass plants, as shown in Figure 6. At the same particle size, the ryegrass height initially increases and then decreases with increasing coal gangue cover thickness (although it remains higher than the control group, CK), reaching its peak at the cover thickness of T3. This further indicates that coal gangue covering has a promoting effect on the height of ryegrass, with the optimal treatment occurring at S2T3. Under the same coal gangue covering treatment, the height of ryegrass continues to increase over time. Specifically, at 42 days of growth, the increase in the height of ryegrass was significantly greater than that at other growth durations. For example, with the S1 particle size, the height of ryegrass at 42 days increased by 7.66 cm, 18.35 cm, and 19.25 cm compared to its height at 35 days. In contrast, the increase in height at 21, 28, 35, 49, 56, and 70 days was all smaller than that at 42 days. This phenomenon is partly due to the growth spurt period observed in ryegrass around the growth stage (35–56 days), similar to the results of Wang Shaoli’s study [34]. At approximately 42 days of growth, a significant change in the plant height of the ryegrass occurred, preceded by a change in the soil pH.

During the ryegrass growth period, the soil pH initially increased and then decreased with the coal gangue coverage treatment. As the coal gangue coverage thickness increased under the same particle size, the soil pH decreased to a value significantly lower than that for the control group. Studies by Wang Qiong et al. and Sun Hairong et al. on the use of coal gangue in soda-affected and alkaline soils, respectively, showed that coal gangue could lower the pH of saline–alkali soils, which is consistent with the results of this study [35,36]. Taking S2 as an example, at 70 days, under the coverage thicknesses T1, T2, and T3, the pH decreased by 0.24, 0.36, and 0.39, respectively, compared to the control group, reaching the optimal value at coverage thickness T2. Under the same coverage treatment, the pH turning points occurred at approximately 28–35 days, after which the ryegrass growth rates increased. The soil pH of the control group did not change significantly, which may be the reason for the lack of noticeable growth in ryegrass. Conversely, the application of coal gangue as a soil cover resulted in a reduction in the pH and alkalinity of the soil, with the pH decreasing to a minimum of 8.49; this falls within the optimal growth range of 5 to 9 for forage crops. This alteration directly influences the availability and state of soil nutrients (such as organic matter, total nitrogen, and urease), enhancing their transformation and effectiveness, and impacting plant growth and development [37,38]. Moreover, coal gangue coverage significantly improved the pore conditions, reduced the bulk density, and increased the water retention and water-holding capacity of soil, promoting soil enhancement. Over time, this enhancement ultimately affected the growth of ryegrass. As the growth of plants in the soil gradually improves, root penetration and other effects will further reduce the soil’s bulk density, enhancing the soil environment. Coal gangue coverage measures had a cumulative effect on the growth of ryegrass through both direct and indirect influences. The pH and alkalinization of soil, either alone or combined with other soil properties, may impact the growth of ryegrass; thus, the specific mechanisms warrant further investigation.

### 2.6. Correlation Analysis

There were highly significant positive correlations between urea (URE) and total nitrogen (TN), and between URE and soil organic matter (SOM) (*p* < 0.001), with correlation coefficients of 0.97 and 0.95, respectively; this indicates a synergistic transport relationship, as depicted in Figure 7. Additionally, there were highly significant positive correlations between SOM and TN, and between the water-filled pore space (WFPS) and available potassium (AK), with correlation coefficients of 0.92 and 0.91, respectively. Furthermore, a significant positive correlation was observed between the water-holding capacity (WHC), URE, and invertase (INV), with correlation coefficients of 0.89 and 0.88. The WHC showed significant positive correlations with SOM, TN, and AK (*p* < 0.01), with correlation coefficients of 0.87, 0.86, and 0.83, respectively. Moreover, significant positive correlations were found between the WFPS and TN, URE, and WHC, with correlation coefficients of 0.85, 0.80, and 0.80. There was also a significant positive correlation between SOM and INV (correlation coefficient of 0.77), as well as between AK and URE, and INV, with correlation coefficients of 0.83 and 0.80. Additionally, there was a significant positive correlation between URE and INV, with a correlation coefficient of 0.87.

The water-holding capacity (WHC) showed a highly significant negative correlation with the exchangeable sodium percentage (ESP) and pH (*p* < 0.001), with correlation coefficients of −0.96 for both. Similarly, pH had a highly significant negative correlation with INV, URE, TN, and AK, with correlation coefficients of −0.95, −0.9, −0.89, and −0.88, respectively. Moreover, ESP exhibited a highly significant negative correlation with URE, TN, and SOM, with correlation coefficients of −0.93, −0.92, and −0.91. Additionally, WFPS had a significant negative correlation with ESP and pH (*p* < 0.01), with correlation coefficients of −0.86 and −0.82. Furthermore, pH had a significant negative correlation with SOM (correlation coefficient of −0.85), while available phosphorus (AP) had a significant negative correlation with INV and alkaline phosphatase (ALP), with correlation coefficients of −0.84 and −0.8. Moreover, AK had a significant negative correlation with URE, TN, and SOM, with a correlation coefficient of −0.8. There was also a significant negative correlation between AK and ESP, with a correlation coefficient of −0.86, and between INV and ESP, with a correlation coefficient of −0.80.

Humidity (H), air-filled water (AFW), and available water (ADW) were significantly positively correlated with URE, TN, SOM, WHC, and WFPS, respectively, (*p* < 0.01). They also showed a positive correlation with AK (*p* < 0.05). Additionally, the urinary fluid weight (UFW) and urinary dry weight (UDW) were significantly positively correlated with TN (*p* < 0.01) and positively correlated with URE (*p* < 0.05). Furthermore, H, AFW, and ADW showed significant negative correlations with ESP (*p* < 0.001), a negative correlation with pH (*p* < 0.05), and a significant negative correlation between UFW, UDW, and ESP (*p* < 0.01).

The correlation studies conclusively established that urease functions as a urea-hydrolyzing enzyme in soil, primarily originating from plants and microbes, and that its activity is intricately linked to the ability of soil to supply nitrogen [39]. The moisture content of soil fosters fungal growth and reproduction, exhibiting positive correlations with the organic matter, high porosity, water content, cover thickness, and larger particle size, and negative correlations with the organic matter content [40].

### 2.7. Principal Component Analysis

Through correlation analysis, it was found that the soil moisture conditions (WHC, WFPS), soil alkalinity (pH, ESP), and nutrient components (URE, SOM, TN) are significantly related to the growth status of ryegrass (H, AFW, ADW, UFW, UDW). To further analyze and discuss the contributions of these seven key influencing factors, namely soil moisture conditions (WHC, WFPS), soil alkalinity (pH, ESP), and nutrient components (URE, SOM, TN), to the growth status of ryegrass, principal component analysis (PCA) was employed; this was used to extract principal components based on the following criteria: eigenvalues greater than one and a cumulative contribution rate of at least 85%. The KMO value from the KMO and Bartlett’s test was found to be 0.895, exceeding the threshold of 0.6 and thus meeting the prerequisite for conducting PCA; this indicated that the data were suitable for PCA studies. Additionally, the data passed Bartlett’s test of sphericity (*p* < 0.05), confirming the appropriateness of the dataset for PCA. The contribution rate of the first principal component was 85.79%, while the contribution rate of the second principal component was 4.67%, as shown in Figure 8. One principal component was extracted, and this had an eigenvalue greater than one. The variance explanation rate of this principal component was 85.792%, resulting in a cumulative variance explanation rate of 85.792%.

The load coefficients, communalities, comprehensive score coefficients, and weights for the saturated water content (WFPS), water-holding capacity (WHC), pH (pH), alkalinity (ESP), urease (URE), organic matter (SOM), and total nitrogen (TN) of soil, as shown in Table 2. The results indicate that the moisture conditions, pH, and nutrient components of soil significantly affect ryegrass growth.

## 3. Discussion

### 3.1. The Impact of Different Coal Gangue Covering Treatments on Soil Moisture Conditions

The soil moisture condition is a crucial physical factor that affects crop roots, thereby influencing their growth and development [41]. The use of coal gangue cover has a significant impact on the moisture conditions of soil [42]. This cover effectively creates a physical barrier between the soil and the external environment. On the one hand, it reduces the contact area between the soil and the atmosphere, thereby limiting the number of pathways via which moisture can evaporate [43]. On the other hand, the presence of coal gangue increases the flow pathways for water, which hinders the infiltration of moisture [44]. Therefore, it is important to consider the combined effects of the thickness and particle size of the coal gangue cover on the suppression of soil moisture evaporation.

Under the dual influence of the coal gangue cover thickness and particle size, the response of soil moisture to changes in particle size is relatively weak, while it shows a significant relationship with variations in cover thickness. A greater cover thickness results in a more pronounced suppression of evaporation, and the soil water storage capacity increases as the thickness of the coal gangue cover increases [45]. This not only reduces the impact of rainfall and soil crusting but also promotes the development of soil pores [46]. Consequently, the porosity of the soil increases, leading to a significant rise in the soil’s saturated moisture content. The mechanism behind this involves alterations in the environment, pH, organic matter, microbial activity, and community composition of the soil due to the presence of coal gangue on the soil’s surface, which in turn modifies the soil’s pore structure.

Additionally, the particle size of coal gangue may influence the movement of water vapor, with the skeletal structure and number of pores playing a decisive role [47]. Coal gangue with a smaller particle size seems to more significantly inhibit evaporation, leading to increased water retention. Therefore, soils treated using coal gangue with a smaller particle size tend to store more moisture. Research by Xu Liangji and others showed that soil treated using coal gangue with a smaller particle had a higher water content than that treated with a larger particle size, consistent with the results of this study [48]. However, when the particle size of coal gangue is relatively small, it exhibits a cohesive structure and significant capillarity. Additionally, the particles of coal gangue are closely packed with the soil, which prevents the soil’s capillary action from being completely disrupted; this results in a lower saturated moisture content. As the particle size of the coal gangue increases, the connectivity improves, leading to a corresponding increase in infiltration rates and, consequently, an increase in the soil’s saturated moisture content. However, this larger particle size also leads to increased evaporation rates. This phenomenon is related to the porosity of the coal gangue cover; larger particles create larger pores that can interrupt capillary action. Nevertheless, moisture in the soil can still evaporate significantly in the form of water vapor, causing the soil’s water-holding capacity to decrease gradually. As the particle size increases, the restriction on soil moisture vapor transport caused by the coal gangue does not significantly improve, leading to the conclusion that an ongoing increase in the coal gangue particle size does not have a noticeable impact on the soil’s saturated moisture content.

### 3.2. Effects of Different Coal Gangue Coverings on Soil Nutrient Element Content

Generally, the nutrient content of the soil increases with the amount of coal gangue used. This is mainly due to the mechanical weathering of coal gangue by plant roots, breaking it into smaller particles and then releasing nutrients into the soil via the enzymatic reactions of soil microorganisms and the acid dissolution reactions of root exudates; this results in a continuous increase in the nutrient content of saline–alkali soil. Coal gangue contains elements such as carbon, nitrogen, phosphorus, and potassium. Underwater infiltration, these nutrients are continuously leached out, altering the soil’s nutrient content [49].

In this study, variance analysis showed that covering the soil with coal gangue significantly increased the total nitrogen, organic matter, and available potassium content of the soil. An increase in the thickness of the coal gangue cover implies that a larger quantity of coal gangue was used; this leads to a higher nutrient content in the leachate, thus enhancing the nutrient levels in the soil. This is due to the release of a substantial number of elements from the mineral structures of coal gangue, such as iron phosphate, aluminum phosphate, and silicates, with potassium being particularly mobile. Potassium in coal gangue is easily leached into the soil with water, potentially increasing the available potassium content of the soil [50,51]. Conversely, the effective phosphorus content significantly decreased as the cover thickness increased, and this reduction was more pronounced when coal gangue with smaller particle sizes was used. The decrease in effective phosphorus in the soil may be attributed to the high concentration of salt ions in the leachate from coal gangue, which raises the calcium ion content of the soil. Consequently, phosphate ions can easily form calcium phosphate precipitates with calcium ions, reducing the availability of phosphorus [52,53]. Additionally, the effective phosphorus content is influenced by the soil’s parent material and microbial activity [54]. The soil’s organic carbon and total nitrogen contents increase as the particle size of the coal gangue decreases. Smaller coal gangue particles have more contact with air and water, allowing nutrients to gradually release and leach into the soil through rainfall, thereby enhancing the soil’s nutrient levels [55,56,57]. The leaching of nutrient elements from coal gangue increases as the particle size decreases. Smaller particles have a larger specific surface area and a greater solid–liquid contact area, facilitating the leaching of nutrient elements [58]. Coverage with coal gangue greatly improves the nutrient content of the soil.

## 4. Material and Methods

### 4.1. Experimental Materials

The soil used in the experiment was collected from the eastern part of the Jingsheng coal mine in the Ningdong Energy and Chemical Industry Base, located in the northwestern area of Ningxia Yinchuan. This area is east of the Yellow River on the edge of the desert, where the eastern monsoon region meets the western arid zone. The coordinates of the location are 106.71° east in longitude and 38.05° north in latitude. The region features a mesothermal arid continental climate characterized by distinct seasons, significant diurnal temperature variations, an annual insolation of 2955 h, and a frost-free period lasting 176 days [59]. The soil type is predominantly sandy, with an annual rainfall of less than 168 mm. The area is prone to strong winds and sandstorms, with over 10 gales above grade 7 recorded. Within a 500 m radius of the Jingsheng coal mine, the surface layer of soil contains approximately 2 cm of coal dust. The soil pH ranges from 9.15 to 9.29, indicating low fertility and an alkalinity level exceeding 20.0%. Soil sampling was conducted at a depth of 100 cm using the profiling method. The soil profile revealed a top layer of sandy soil, beneath which lay a coarse soil layer with a high clay content (86.62% sand, 12.61% silt, 0.97% clay). Subsequent soil processing involved the removal of plant roots, gravel, and other debris. The soil samples were then air-dried, ground, sieved through a 1 mm mesh, and analyzed for basic physical and chemical soil indicators. The remaining soil was allocated for the potting experiments [60]. The physical properties of the test soil are listed in Table 3, and the soil nutrient elements are presented in Table 4.

Coal gangue was collected from Ningdong Town, Lingwu City (38°05′ N, 106°43′ E). Unweathered gangue samples were extracted from the gangue pile and brought back to the laboratory, crushed with a steel hammer, and air-dried. The gangue was thoroughly mixed and divided into the following three particle size groups using stainless steel sieves: 0–0.5 cm, 0.5–1 cm, and 1–2 cm. The gangue predominantly comprised gray–black marl shale sandstone, with mud calcite, illite, kaolinite, feldspar, and quartz as the main mineral components. The gangue contained approximately 15–20% carbonaceous matter, 60–70% marl-like components, and 5–10% other minerals [61]. The heavy metal content was well below the national limits, making it suitable for use as a safe covering material. To prevent the influence of gangue weathering on its characteristics, only fresh, unweathered, and black-colored gangue samples were used in this experiment [13]. Details of the physical and chemical characteristics of the gangue are presented in Table 5, while the heavy metal detection results are summarized in Table 6.

The crop variety chosen was ryegrass obtained from XiBei Agriculture, Forestry, and Animal Husbandry Ecological Science and Technology Co, Ltd. in Yinchuan, China. Only intact, uniform, and pest-free seeds were used in this study. The crop selection methods are described in Section 2.2, Experimental Methods.

### 4.2. Experimental Methods

The experiment commenced on 26 March 2024, at the Key Laboratory of Arid Zone Resource Evaluation and Environmental Regulation of Ningxia University (38°30′ N, 106°19′ E), and lasted 70 days. Sieved soil samples, spread to a thickness of 1–2 cm in drying trays, were placed in a well-ventilated area [62,63]. Once dry, 4 kg samples were added to planting buckets, maintaining an initial moisture content of 3.44%. Each bucket had three 1 cm drainage holes, covered with gauze to prevent soil leakage. To assess the effects of cover thickness (4 cm, 8 cm, and 12 cm) on the physicochemical properties and microbial biomass of soil, nine treatments were designed: three thicknesses (T1, T2, T3) and three particle sizes (0–0.5 cm, 0.5–1 cm, and 1–2 cm), plus an uncovered control (CK); this resulted in 30 experimental units with three replicates [64,65,66]. For ryegrass germination, seeds were sun-dried for 2 h, soaked for 12 h, and only those that sank were sown, with 50 seeds per pot. The pots were randomly relocated every 7 days, totaling 10 changes. The new positions, randomly assigned, were recorded for comparison; in addition, relocations were performed carefully to avoid root damage, and primarily in the evening to minimize light disruption [67]. Watering was performed every 7 days, starting with a uniform pre-watering of 1000 mL per pot, followed by 800 mL per pot in subsequent irrigations, ensuring that there was adequate moisture without waterlogging [68]. The water volumes were accurately measured to ensure the even distribution of moisture. The temperature was kept at 25 °C, and periodic adjustments to the position of the pot minimized the effects of light exposure, with weekly observations recorded [69].

Ryegrass typically emerged around 10 days post-sowing and matured in approximately 70 days. The plant height and soil pH were recorded every 7 days. At the end of the 70 days, the physicochemical properties of the soil, its nutrient elements, and the ryegrass biomass were assessed. Soil samples were taken from a 0–20 cm depth using a ring knife, yielding 30 samples free of roots, animal remains, and stones. The samples were sieved through a 2 mm mesh and divided into two parts for storage: one for physicochemical analysis and the other for determining the enzyme activity of the soil [59].

The physicochemical properties of the soil were assessed according to Soil Agrochemical Analysis [70]. The bulk density of the soil was measured using the ring knife method (100 cm^3^) [71]. The saturated water content and holding capacity of the field were determined via the weight method, with pH analyzed via the electrode method [72,73]. The organic matter was oxidized with potassium dichromate, and total nitrogen was evaluated using the Kjeldahl method [74,75]. Effective phosphorus was extracted via sodium bicarbonate leaching, and fast-acting potassium was leached with NH_4_OAc; this was measured using flame photometry [76,77]. The total salt content was assessed via conductivity, and exchangeable Na^+^ was determined using the ammonium acetate–ammonium hydroxide method (LY/T148-1999) [78,79]. The exchangeable sodium percentage (ESP) was calculated according to the exchangeable Na^+^ among cations [80]. The activity of enzymes was measured using methods from Soil Enzymes and Their Research Methods [81]. Soil urease was assessed through indophenol blue colorimetry, alkaline phosphatase using disodium benzene phosphate, and sucrase using 3,5-dinitrosalicylic acid [82,83,84].

The ryegrass height was measured with a straightedge (scale = 1 mm), and the biomass was determined using a balance (precision 1:10,000). Upon maturity, the above-ground parts were harvested at soil level and weighed to determine their fresh weight. The roots were extracted, washed, and weighed, with a portion heated at 105 °C for 30 min, then dried at 70 °C for 72 h; their weights were recorded until constant [85]. The above-ground and below-ground dry weights were recorded at maturity, with a similar treatment used for roots to assess the total dry weight for the maturation period. The experimental pre-treatment and experimental design are shown in Figure 9.

### 4.3. Data Processing

The data analysis employed ANOVA and LSD to evaluate the differences between factors across various treatments, considering a significance of *p* < 0.05. Correlation analysis was performed to determine the correlation coefficients between the factors. Principal component analysis plots were created and optimization was performed using R2020 and Illustrator2020. Statistical analysis was performed using SPSS 29.0 software (IBM Corp, Armonk, NY, USA), and graphical representations were created with OriginPro 2022 software (OriginLab software Inc., Northampton, MA, USA).

Principal component analysis (PCA) is a dimensionality reduction method in which variables are selected based on the variance explained by each indicator and the cumulative contribution until the selected variables represent the information of the original indicator. Subsequently, the combined score of each principal component is calculated [86].

(1)The formula for calculating each principal component score is as follows:(1)$$\left\{\begin{array}{c} F_{1}=Z_{11}X_{1}+Z_{12}X_{2}+\cdots +Z_{1j}X_{j}\\ F_{2}=Z_{21}X_{1}+Z_{22}X_{2}+\cdots +Z_{2j}X_{j}\\ \vdots \\ F_{n}=Z_{n1}X_{1}+Z_{n2}X_{2}+\cdots +Z_{nj}X_{j} \end{array}\right.$$(2)The data for the calculation of the combined score are as follows:(2)$$F=\frac{\left(b_{1}F_{1}+b_{2}F_{2}+\cdots +b_{n}F_{n}\right)}{100}$$
where F is the composite score; $b_{1}$, $b_{2}$, ⋯, and $b_{n}$ are the variance contribution rates; and $F_{1}$, $F_{2}$, ⋯, $F_{n}$ are the scores of each principal component.

## 5. Conclusions

This study investigated the effects of coal gangue cover on the physicochemical properties of saline–alkali soil, the enzyme activity of the soil, and the growth of ryegrass. We found that when the size of the coal gangue particles ranged between 0 and 1 cm and the cover thickness ranged from 8 to 12 cm, the soil moisture was effectively increased, the soil alkalinity was reduced, and the soil’s nutrient content was enhanced (including soil organic matter, total nitrogen content, and soil urease activity), thus promoting the growth of ryegrass. Among these, a 12 cm cover with a particle size of 0.5–1 cm had the most positive effect on ryegrass growth. The results provide a theoretical basis for improving the water-holding capacity of saline–alkali soil, reducing its alkalinity, enhancing its nutrient content, and improving crop growth indicators.

To strengthen the ameliorating effects of the coal gangue cover on saline–alkali soil and to further apply coal gangue cover technology in mining areas, the incorporation of soil amendments (like vermicompost, biochar, etc.) or the use of combined covers with other materials should be considered. Exploring the synergy of coal gangue with other covers by examining factors such as soil properties (moisture, structure, nutrients, and microbial activities) and crop growth. Further exploration of the impacts and mechanisms of coal gangue, in conjunction with other covers on the improvement of saline–alkali soil.

## Figures and Tables

**Figure 1 plants-13-03419-f001:**
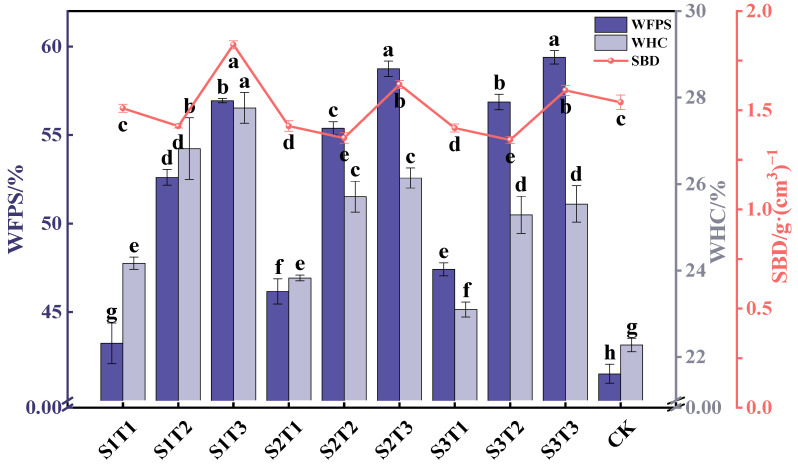
The impact of various treatments on the saturated water content, water-holding capacity, and bulk density of soil. Dotted line plots and histograms depict the mean values with standard deviations (*n* = 3). Note: WFPS represents the saturated water content of soil, WHC denotes the water-holding capacity of soil, and SBD refers to the bulk density of soil. Different letters indicated significant differences among different treatments (*p* < 0.05) by LSD test.

**Figure 2 plants-13-03419-f002:**
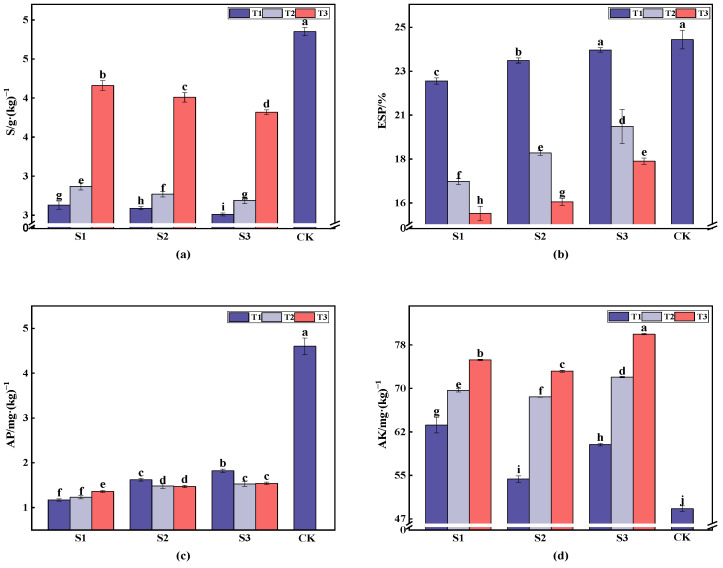
The impact of various treatments on the total salt, alkalinity, effective phosphorus, and fast-acting potassium content of soil. The histograms represent mean ± standard deviation (*n* = 3). Specifically, (**a**) shows the total salt content of the soil under different treatments, (**b**) depicts the alkalinity of soil under different treatments, (**c**) shows the effective phosphorus content of the soil under different treatments, and (**d**) presents the fast-acting potassium content of the soil under various treatments. Different letters indicated significant differences among different treatments (*p* < 0.05) by LSD test.

**Figure 3 plants-13-03419-f003:**
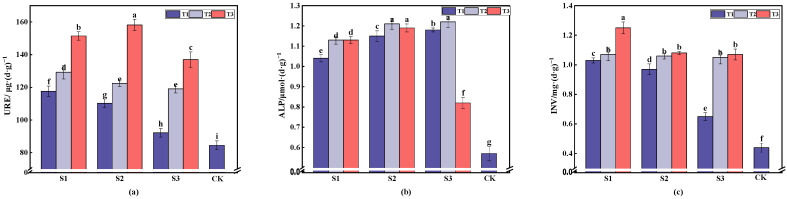
The impact of various treatments on the urease, alkaline phosphatase, and sucrase in the soil. The histograms depict the mean values along with standard deviations (*n* = 3). Furthermore, (**a**) shows the urease content of the soil under different treatments, (**b**) shows the alkaline phosphatase content of the soil under different treatments, and (**c**) presents the sucrase content of the soil under different treatments. Different letters indicated significant differences among different treatments (*p* < 0.05) by LSD test.

**Figure 4 plants-13-03419-f004:**
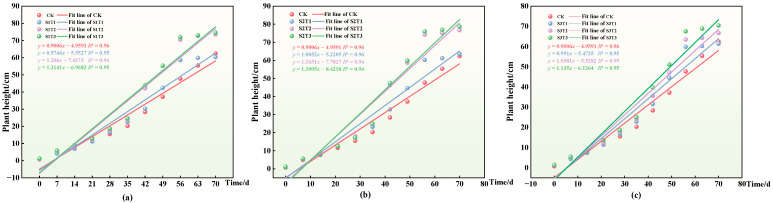
The application of the logistic growth model for fitting the growth trajectory of ryegrass plant height from 0 to 70 days under various treatments. Specifically, the figure shows the following: (**a**) linear fitting for the S1 grain size with varying cover thicknesses, (**b**) linear fitting for the S2 grain size with different cover thicknesses concerning the ryegrass plant height, and (**c**) linear fitting for the S3 grain size with differing cover thicknesses regarding ryegrass plant height.

**Figure 5 plants-13-03419-f005:**
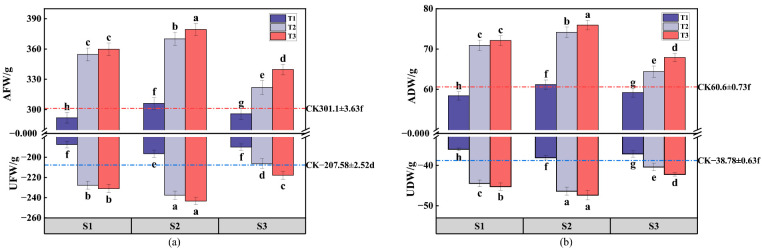
The impact of various treatments on the fresh and dry weights of ryegrass, both above and below ground. The dotted line plots and histograms depict the mean values along with the standard deviations (*n* = 3). Specifically, (**a**) represents the fresh weight of aboveground (AFW) and belowground (UFW) ryegrass under different treatments, while (**b**) represents the corresponding dry weights (ADW for above ground and UDW for below ground). Different letters indicated significant differences among different treatments (*p* < 0.05) by LSD test.

**Figure 6 plants-13-03419-f006:**
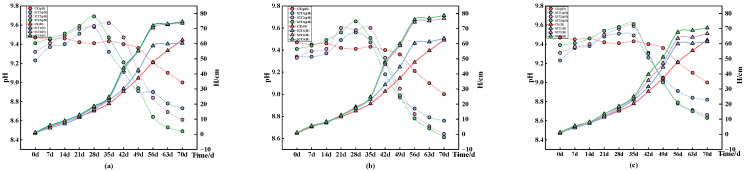
The changes in soil pH and ryegrass plant height with growth time under different treatments. Dotted line plots represent mean ± standard deviation (*n* = 3). Note: (**a**) changes in soil pH and ryegrass plant height under different thicknesses and S1 particle size coverage from 0 to 70 days, with H representing the ryegrass plant height; (**b**) changes in soil pH and ryegrass under different thicknesses and S2 particle size coverage from 0 to 70 days; (**c**) changes in soil pH and ryegrass plant height under different thicknesses and S3 particle size coverage from 0 to 70 days.

**Figure 7 plants-13-03419-f007:**
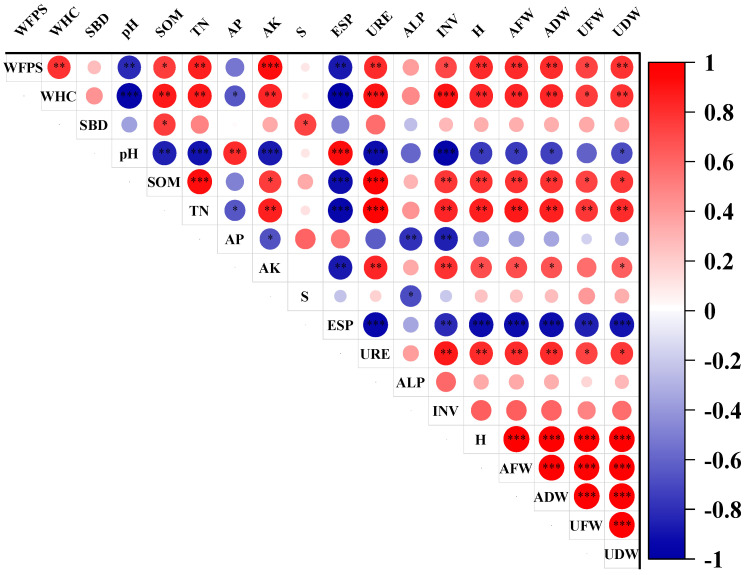
Correlation analysis of physicochemical properties of soil and ryegrass indicators under different treatments. *** denotes highly significant correlation at the 0.001 level, ** denotes significant correlation at the 0.01 level, * denotes correlation at the 0.05 level; sample size *n* = 54. WFPS, soil saturated water content; WHC, soil water-holding capacity; SBD, soil bulk density; pH, soil acidity and alkalinity; SOM, soil organic matter; TN, soil total nitrogen; AP, soil effective phosphorus; AK, soil fast-acting potassium; S, soil total salt; ESP, soil alkalinity; URE, soil urease; ALP, soil alkaline phosphatase; INV, soil sucrase; H, ryegrass plant height; AFW, above-ground fresh weight of ryegrass; ADW, above-ground fresh weight of ryegrass; UFW, above-ground dry weight of ryegrass; UDW, below-ground dry weight of ryegrass. Note: The blue numbers in the figure indicate the negative correlation between the variables, and the shade of the color indicates the strength of the negative correlation; the red numbers indicate the positive correlation between the variables, and the shade of the color indicates the strength of the positive correlation.

**Figure 8 plants-13-03419-f008:**
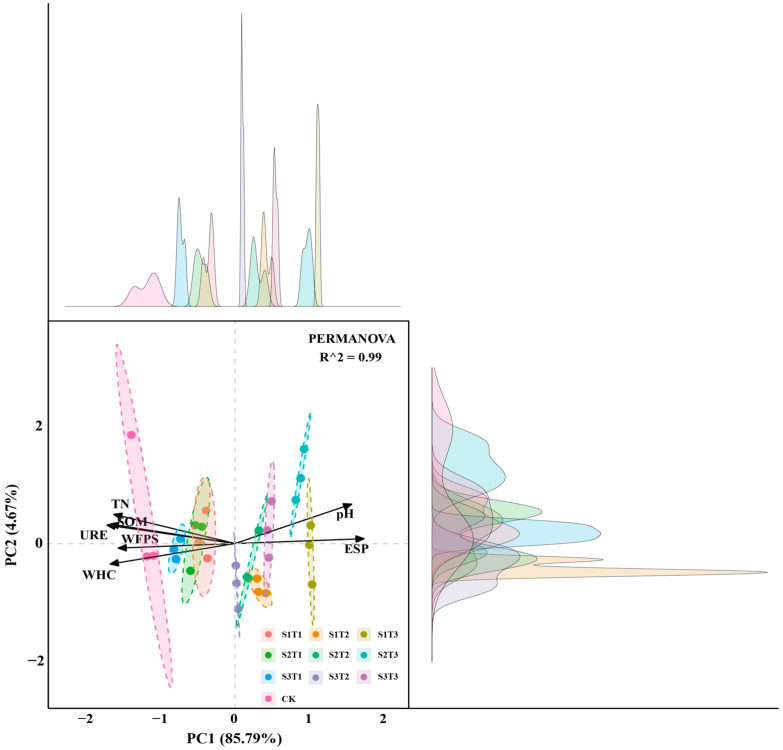
Principal component analysis of moisture conditions, pH, and nutrients of soil under different treatments.

**Figure 9 plants-13-03419-f009:**
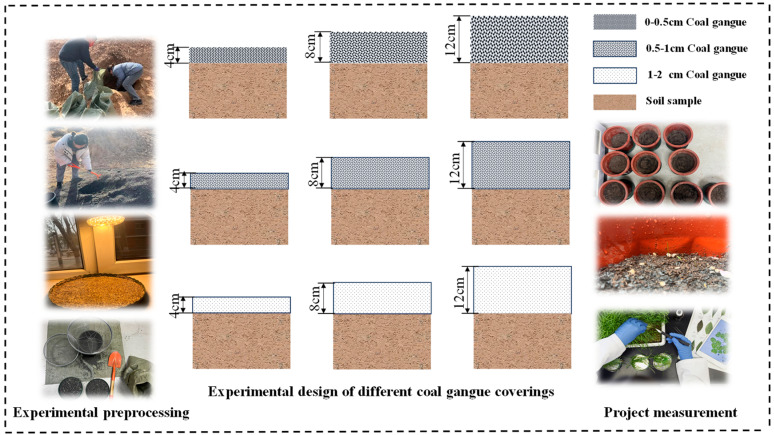
Diagram of the experimental design.

**Table 1 plants-13-03419-t001:** Effect of various treatments on pH, organic matter, and total nitrogen. Different letters indicated significant differences among different treatments (*p* < 0.05) by LSD test.

Treatment	pH	SOM/(mg/kg)	TN/(mg/kg)
S1T1	8.73 b	11.25 g	0.17 d
S1T2	8.61 d	13.46 d	0.19 c
S1T3	8.49 a	21.54 a	0.22 a
S2T1	8.76 b	10.87 g	0.16 e
S2T2	8.64 c	12.89 e	0.20 b
S2T3	8.61 d	19.57 b	0.24 a
S3T1	8.82 b	8.94 h	0.15 e
S3T2	8.66 c	11.81 f	0.17 d
S3T3	8.63 d	15.34 c	0.21 b
CK	9.00 a	7.90 i	0.12 f

**Table 2 plants-13-03419-t002:** The results of the principal component analysis, total variance interpretation of indicators and component matrix.

Analyzed Item	Load Factor	Communality	Principal Component 1 (85.79%)	Comprehensive Score Coefficient	Weight
WFPS	0.879	0.772	0.3585	0.3585	13.56%
WHC	0.938	0.880	0.3829	0.3829	14.48%
pH	−0.878	0.771	0.3584	−0.3584	13.56%
SOM	0.938	0.879	0.3826	0.3826	14.47%
TN	0.912	0.832	0.3722	0.3722	14.08%
ESP	−0.971	0.943	0.3963	−0.3963	14.99%
URE	0.963	0.927	0.3930	0.3930	14.86%

**Table 3 plants-13-03419-t003:** Physical and chemical characteristics of the test soil.

Data Source	Initial Soil Moisture Content (%)	Bulk Density (g/cm^3^)	pH	Soil Alkalinity (%)	Soil Salt Content (g/kg)	Organic Matter (g/kg)
Soil sample	3.44	1.59	9.29	25.94	2.10	396.00
Reference value for saline soils *		-	>7.00	>5.00	>0.20	-

Note: “-” indicates no reference value; * data from the literature.

**Table 4 plants-13-03419-t004:** Nutrient element contents of test soils.

Available Nitrogen (mg/kg)	Available Phosphorus (mg/kg)	Available Potassium (mg/kg)	Soil Urease μg/(d·g)	Soil Alkaline Phosphatase μmol/(d·g)	Soil Saccharase mg/(d·g)
39.46	4.03	94.71	83.12	0.56	0.42

**Table 5 plants-13-03419-t005:** Main composition indexes of the gangue used for testing.

Data Source	pH	Initial Soil Moisture Content (%)	Bulk Density (g/cm^3^)
Coal gangue	9.70	1.43	1.61

**Table 6 plants-13-03419-t006:** Heavy metal detection results for the gangue used for testing.

Data Source	Cd/(mg/kg)	Pb/(mg/kg)	Cr/(mg/kg)	Hg/(mg/kg)	As/(mg/kg)	Cu/(mg/kg)	Zn/(mg/kg)	Ni/(mg/kg)
Coal gangue	0.274	69.1	64.6	0.199	1.35	38.1	108	48.3
Screening value	<0.6	170	250	3.4	25	100	190	300

## Data Availability

Data are contained within the article.

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
