# Peer review of "Improving Saline–Alkaline Soil and Ryegrass Growth with Coal Gangue Treatments"

_plants, 2024, doi:10.3390/plants13233419_

Round 1
Reviewer 1 Report
Comments and Suggestions for Authors
This manuscript investigates the innovative use of coal gangue as a covering material to improve saline-alkali soil properties and ryegrass growth. It provides valuable insights into how varying particle sizes and thicknesses influence soil moisture, enzyme activity, and nutrient content. The study employs robust experimental setups and statistical analyses to validate its findings. Its potential applications in soil restoration and sustainable agriculture make it a significant contribution to the field.
Major Comments:
· The objectives in the Introduction (Lines 76–83) are not clearly stated. While the study aims to assess the effects of coal gangue on saline-alkali soils, the hypothesis and specific goals are unclear.
· The methods section (Lines 135–152) lacks an explanation for selecting specific coal gangue thicknesses and particle sizes.
· Correct line 103-105; 121-122; 194; 224; 332; 394; 434; : Error! Reference source not found.,
· While the results on soil properties and enzyme activities are detailed (Lines 218–426), the discussion (Lines 580–666) does not adequately address the mechanisms driving the observed effects. Referencing prior studies or proposing hypotheses would be more appropriate
· A similar trends in soil properties and enzyme activities are described repetitively across the results section (Lines 218–426). Consolidating these findings would enhance clarity and readability.
Minor Comments:
· Line 18: "regress growth" should be "ryegrass growth”.
· Terms like "WHC" and "WFPS" (Lines 218–244) are repeatedly defined. Define them once and ensure consistent use.
· Minor grammar and abbreviations should be checked across the manuscript.
Comments on the Quality of English LanguageThe English could be improved to more clearly express the research.
Reviewer 2 Report
Comments and Suggestions for Authors
I have reviewed the manuscript, and it needs extensive revision, both in terms of its contents and formatting.
Firstly, follow my comments in the attached file.
Secondly, you should follow the MDPI manuscript preparation instructions.
Importantly, you have not mentioned any figure or table number in the main file. This should correspond to and be reflected in the main Text.
Also, the reference section arrangements should be rearranged as per MDPI policy, as you should mention all the authors instead of etc.

Extensive english editing is required.
Round 2
Reviewer 2 Report
Comments and Suggestions for Authors
Thanks for considering my suggestions. The manuscript improved a lot now. For me, I'm happy with the revised version.
Just need to consider some minor revisions. Please see the attached file for your consideration.

Minor editing of English is required.
